# Initial report on polar mesospheric cloud observations by Himawari-8

Takuo T. Tsuda[1], Yuta Hozumi[1], Kento Kawaura[1], Keisuke Hosokawa[1], Hidehiko Suzuki[2], and Takuji Nakamura[3,4]

[1]Department of Computer and Network Engineering, The University of Electro-Communications, Chofu, Japan.
[2]Department of Physics, Meiji University, Kawasaki, Japan.
[3]National Institute of Polar Research (NIPR), Tachikawa, Japan.
[4]Department of Polar Science, Graduate University for Advanced Studies (SOKENDAI), Tachikawa, Japan.

**Correspondence:** Takuo T. Tsuda (takuo.tsuda@uec.ac.jp)

**Abstract.** We provide an initial report on polar mesospheric cloud (PMC) observations by the Japanese Geostationary-Earth-Orbit (GEO) meteorological satellite Himawari-8. Heights of the observed PMCs were estimated to be 80–82 km. Observed PMCs were active only during summertime in both the northern and southern polar regions. These observations are consistent with known PMC behavior. From its almost fixed location relative to the Earth, Himawari-8 is capable to continuously monitoring PMC every 10 min with three visible bands: blue (0.47 $\mu$m), green (0.51 $\mu$m), and red (0.64 $\mu$m). Thus, Himawari-8 would contribute to the PMC research in the near future.

## 1 Introduction

Polar mesospheric clouds (PMCs) or noctilucent clouds (NLCs) consist of water-ice particles, which can be produced in summer at the mesopause region, mainly at high latitudes. The first report on PMCs was made by Leslie (1885). Since then, various methods have been used to perform PMC observations. Optical observations by ground-based cameras, imagers or lidars are often limited by weather conditions, because a clear sky is required for such observations. Hence, satellite observations from space are valuable for more continuous observations, which enable significant systematic data coverage. Such systematic data coverage would be of benefit, for example, for monitoring long-term PMC activity, which may be related to global change (cf. Thomas, 1996; von Zahn, 2003) because water-ice particle production can be enhanced by $CO_2$ cooling and $H_2O$ increase, which may be induced by $CO_2$ and $CH_4$ increases (cf. Roble and Dickinson, 1989).

A comprehensive review of PMC observations from satellites can be found in DeLand et al. (2006). In addition, the Aeronomy of Ice in the Mesosphere (AIM) satellite has been in operation, making PMC observations since 2007 (Russell et al., 2009). These observations include both limb- and nadir-viewing from low-Earth-orbit (LEO) satellites. By contrast, there are only few reports of PMC observations by limb-viewing Geostationary-Earth-Orbit (GEO) satellites (Gadsden, 2000a, b, 2001; Proud, 2015). The first PMC observations from a GEO satellite were reported using images by Meteosat First Generation (MFG) (Gadsden, 2000a, b, 2001), and their PMC images had $\sim$2.5-km spatial resolution in a single visible band. Subse-

quently, Proud (2015) extended such observations to Meteosat Second Generation (MSG), and he reported PMC observations using ∼1-km spatial resolution images in a single visible band. This kind of GEO satellite can produce full-disk images including the Earth's limb, which would provide valuable opportunities for PMC observations by continuous limb-viewing from its almost fixed location relative to the Earth.

In the present paper, we make an initial report on PMC observations from Himawari-8, the Japanese GEO meteorological satellite. Our PMC images from Himawari-8 have ∼1-km spatial resolution in three visible bands. Japanese GEO meteorological satellites have a long history from 1977 (Himawari-1) to the present (Himawari-8). However, there was no PMC report from Japanese GEO satellite observations before this work. Therefore, in the present paper we examine basic features in PMC emissions observed by Himawari-8 and compare those with typical PMC characteristics, as a first step for our PMC research using Himawari-8 data.

## 2   Data

Himawari-8 is the Japanese GEO meteorological satellite (Bessho et al., 2016), that was successfully launched in October 2014. It has 16 observation bands, including three visible bands: blue (0.47 $\mu$m), green (0.51 $\mu$m), and red (0.64 $\mu$m). In the initial survey for PMCs, we used full-disk images in Portable Network Graphics (PNG) format, generated from the level-1a data, Himawari Standard Data (HSD). The PNG full-disk image is a true-color image, i.e., a composite of the three visible bands. Each color has a 8-bit resolution (i.e., values ranging from 0 to 255), describing emission intensities from 0 to 641.5092 W m$^{-2}$ sr$^{-1}$ $\mu$m$^{-1}$ for the blue band, from 0 to 601.9766 W m$^{-2}$ sr$^{-1}$ $\mu$m$^{-1}$ for the green band, and from 0 to 519.3457 W m$^{-2}$ sr$^{-1}$ $\mu$m$^{-1}$ for the red band. The color value has a linear relation with the emission intensity for each band. The PNG full-disk image has a spatial resolution of ∼1 km and is obtained every 10 min. The geometric accuracy of the images is typically less than 0.6 km, i.e., less than the ∼1-km spatial resolution. More detailed information for the PNG images can be found in Bessho et al. (2016). For the present survey, we collected a year of PNG images for 2016, and focused our attention on the Earth's limb region, namely the middle and upper atmospheric regions.

## 3   Results and discussion

Figure 1 shows an example of PMC emissions observed at 21:00 UT on 9 July 2016 in the northern high latitudes. It is difficult to see any emissions in the limb region in the original true-color image (see Figure 1a), but the 50× enhanced image makes it obvious that clear emissions exist in the limb region (see Figure 1b). The appearance of these emissions is similar to that of PMC emissions in previous reports (Gadsden, 2000a, b, 2001; Proud, 2015). Here, we calculated tangential points in each line-of-sight (LOS) direction (i.e., for each pixel in the image). The image is described by the normalized geostationary projection,

so the pixel corresponds to the LOS angle. From each LOS angle, we derived each vector along each LOS direction. Then, we considered intersections between each vector and the Earth-like ellipsoid of eccentricity defined by World Geodetic System 1984 (WGS84). The equation for the intersections is expressed as a quadratic equation. Hence, if there is only a single solution of the equation, there is only a single intersection that corresponds to the tangential point. Thus, we solved the equation by changing the radius of the ellipsoid to produce only a single solution. Thus, we obtained information for the tangential points for each pixel in the image. The heights and latitudes of tangential points are overlaid in Figure 1c. As shown in Figure 1c, the typical height of the emissions was about 80 km.

For further details, Figure 2 shows height-latitude distributions of emission intensities in the three visible bands. The emissions were mainly located at 80–82 km height at latitudes of 78–81°N. This emission height is consistent with typical reported PMC heights of 82–83 km (cf. DeLand et al., 2006). It should be noted that the tangential height may be an underestimation of the actual emission height because emissions may be coming from not only the tangential point but also foreground or background points. Our observations also show oblique, step-like structures with a spatial scale of ∼1 km. These artifactual structures are because of the height resolution and the mapping calculation of the tangential point. Before application of the mapping calculation, images show step-like structures that are not oblique but horizontal, with a spatial scale of ∼1 km, because of ∼1-km height or spatial resolution. These artifactual structures are because of a limitation of the height resolution. Such horizontal, step-like structures then become oblique, step-like structures through the mapping calculation from the pixel coordinate in the original images to the height-latitude coordinate in the tangential points.

We confirm that the PMC-like emission layer has a wavelike structure in the height-latitude cross section. For example, the heights of the layer were 80–81 km at ∼81°N, 81–82 km at ∼80°N, ∼81 km at ∼79°N, and 81–82 km at ∼78°N. We attribute these fluctuations to atmospheric waves. Because the latitude range of 79–81°N corresponds to about a distance of ∼700 km over the Arctic ocean (see Figure 3), the wavelength of the wavelike structure can be estimated to be ∼700 km. Such PMC structures can be observed by several methods using observations from LEO satellites such as the Cloud Imaging and Particle Size experiment (CIPS) onboard AIM and the Optical Spectrograph and InfraRed Imager System (OSIRIS) onboard Odin. For example, AIM/CIPS can provide PMC nadir imaging, which is a powerful tool to observe horizontal information for PMC structures (e.g., Chandran et al., 2010; Yue et al., 2014; Zhao et al., 2015). In addition, tomographic techniques using AIM/CIPS (Hart et al., 2018) and Odin/OSIRIS (Hultgren et al., 2013) can provide horizontal and vertical information for PMC structures. By contrast, Himawari-8 observation would feature high-cadence and wide limb-viewing PMC observation from its almost fixed location relative to the Earth. The Himawari-8 full-disk image can be obtained every 10 min, and its FOV coverage can be several thousands of kilometers (see Figure 3). These features would provide valuable data, which can be complementary to PMC data from the LEO satellites.

Emission intensity was the strongest in the blue band, and the weakest in the red band. This result can be explained by Rayleigh or Mie scattering of sunlight by water-ice particles (i.e., PMCs). According to the Mie theory (cf. Bohren and Huff-

man, 2007), Rayleigh scattering is Mie scattering in the limit where particle size is much smaller than the wavelength. In the Rayleigh scattering region, the scattering cross section at a fixed wavelength decreases with decreasing particle size, and the scattering cross section at a fixed particle size decreases with increasing wavelength. Most PMC particles are understood to have radii of 20–60 nm, along with a very small number of ∼200 nm particles (cf. DeLand et al., 2006). Such particle size is smaller than the visible observation wavelengths: blue (0.47 $\mu$m), green (0.51 $\mu$m), and red (0.64 $\mu$m). Hence, the scattering would be close to Rayleigh scattering, whereas it may not be the pure Rayleigh scattering. Therefore, the predominance of such small particles (20–60 nm) would account for the higher scattering intensities or stronger emission intensities at shorter wavelengths. Although additional quantitative evaluation would be helpful, the inferential framework described above would imply that the three visible images obtained by Himawari-8 may contain information on the size of PMC particles. In addition, Himawari-8 observation can cover an entire day of local time (i.e., 00:00–24:00 JST), which means that emission data at almost all scattering angles (i.e., 0–360°) are available. Such information may be useful for particle size investigation, considering the wavelength dependence and the angle distribution in the Rayleigh to Mie scattering region. However, this concern is beyond the scope of the present paper.

To investigate seasonal variation of PMC emissions in 2016, we calculated total pixel values in specific regions (i.e., sum of pixel values in localized pixels). We set two regions; one is a region at heights of 70–90 km and latitudes of 60–90°N for the northern polar region (see Figure 4a), whereas the other is a region at heights of 70–90 km and latitudes of 60–90°S for the southern polar region (see Figure 4b). The height range of 70–90 km covers typical PMC occurrence heights, and the latitude range of 60–90° covers typical PMC occurrence latitudes. It should be noted that the upper limit of the latitude range is actually ∼81° (see Figure 3), that is, the highest latitude of the tangential points when the height range is set to 70–90 km. Seasonal variation in the total pixel values is shown in Figure 4. To remove within-day variation, data in Figure 4 include only data at a single local time each day, 06:00 JST (21:00 UT). At that local time, 06:00 JST, the LOS of Himawari-8 is almost perpendicular to the sunward direction. This configuration would be beneficial, providing solar illumination to some extent while minimizing sun-induced noise, such as the stray light of direct sunlight, which can be a problem close to local midnight (00:00 JST) (cf. Proud, 2015). As shown in Figures 4a and 4b, PMC-like emissions, expressed as total pixel values, were active only during local summer months. The observed active periods would be similar to the typical PMC active period, from ∼20 days before summer solstice to ∼60 days after summer solstice (cf. DeLand et al., 2006).

As discussed in the above text, the heights and seasonal variations of the PMC emissions are consistent with the general characteristics of PMCs. These results suggest that the peculiar emissions observed by Himawari-8 are indeed PMCs. We further suggest that the availability of imaging in three visible bands constitutes a particular advantage of Himawari-8 for PMC study. This capability may provide valuable opportunities, for example, for obtaining information on the size of PMC particles. In addition, collaborations between Himawari-8 and LEO satellites such as AIM would allow a synergy of complementary capabilities. In particular, high-time resolution data (imaging every 10 min) from Himawari-8, when combined with data from

LEO satellites, can contribute to PMC research in the near future, e.g., diurnal PMC variation.

## 4 Summary

In this paper, we introduced new PMC observations by Himawari-8. These observations concerning PMC height and seasonal PMC activity are consistent with previously reported PMC characteristics (cf. DeLand et al., 2006). Although these are not new scientific findings, they serve as a brief but essential validation of Himawari-8 observational data as PMC emissions. The detection of emissions consistent with PMC emissions demonstrates that Himawari-8 is capable of PMC observations. Himawari-8 can indeed provide data on new aspects of PMCs, and it can provide valuable opportunities for new PMC research in the near future. In particular, Himawari-8 PMC observations would have the following advantages: (1) high-time resolution (imaging every 10 min); (2) high-spatial resolution (one pixel every ∼1 km) in the height-latitude (or height-horizon) cross-section; (3) three visible bands (blue: 0.47 $\mu$m, green: 0.51 $\mu$m, and red: 0.64 $\mu$m); (4) continuous monitoring from its almost fixed location relative to the Earth.

*Data availability.* The Himawari-8 data, provided by the Meteorological Satellite Center of the Japan Meteorological Agency (JMA), are available from the National Institute of Information and Communications Technology (NICT) Science Cloud.

*Competing interests.* The authors declare that they have no conflict of interest.

*Acknowledgements.* The authors are grateful to Mr. Ryota Yamamoto for providing information on PMC emissions in full-disk images from Himawari-8; this information enabled this work. Himawari-8 data were provided by the Meteorological Satellite Center of the Japan Meteorological Agency (JMA) through the National Institute of Information and Communications Technology (NICT) Science Cloud. This work was supported in part by MEXT/JSPS KAKENHI Grants, JP15H05815, JP16H01171, JP16H06021, and JP17H02968, by the National Institute of Polar Research (NIPR) through General Collaboration Project no. 28-2, and by the joint research program of the Institute for Space-Earth Environmental Research (ISEE), Nagoya University.

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

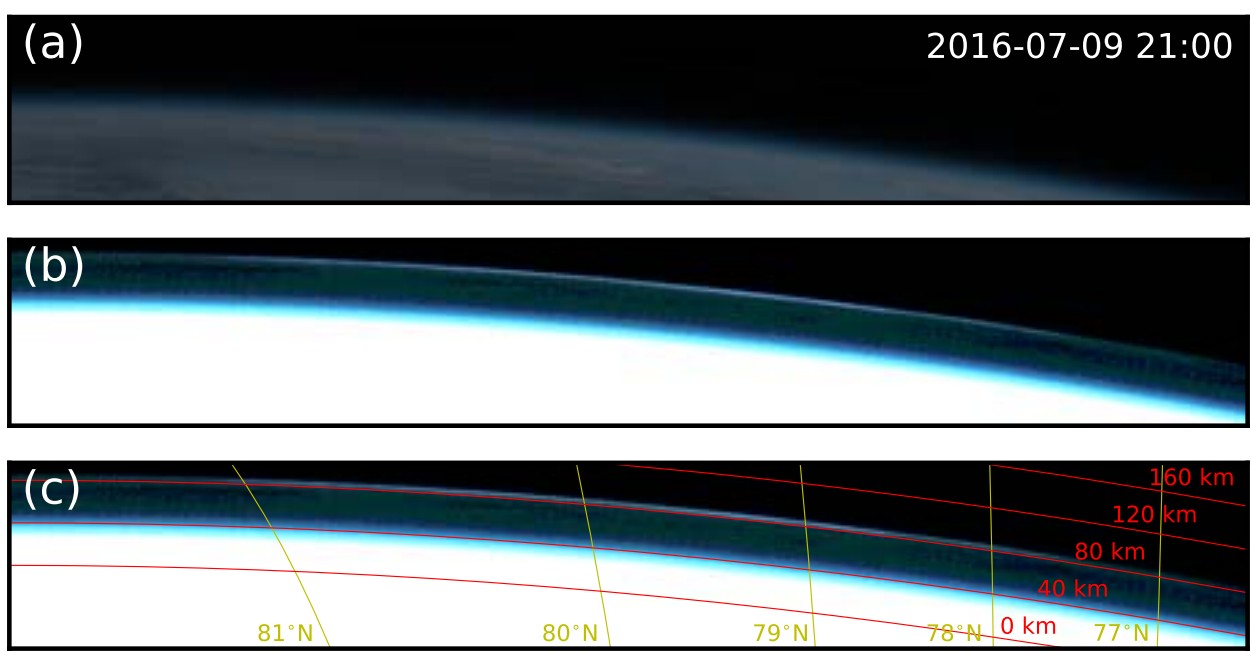

**Figure 1.** (a) An original true-color image (composite of the three visible bands) around northern high latitudes at 21:00 UT on 9 July 2016. (b) Same as Figure 1a, but the color scale is 50× enhanced. (c) Same as Figure 1b, but latitudes and heights of the tangential points are overlaid.

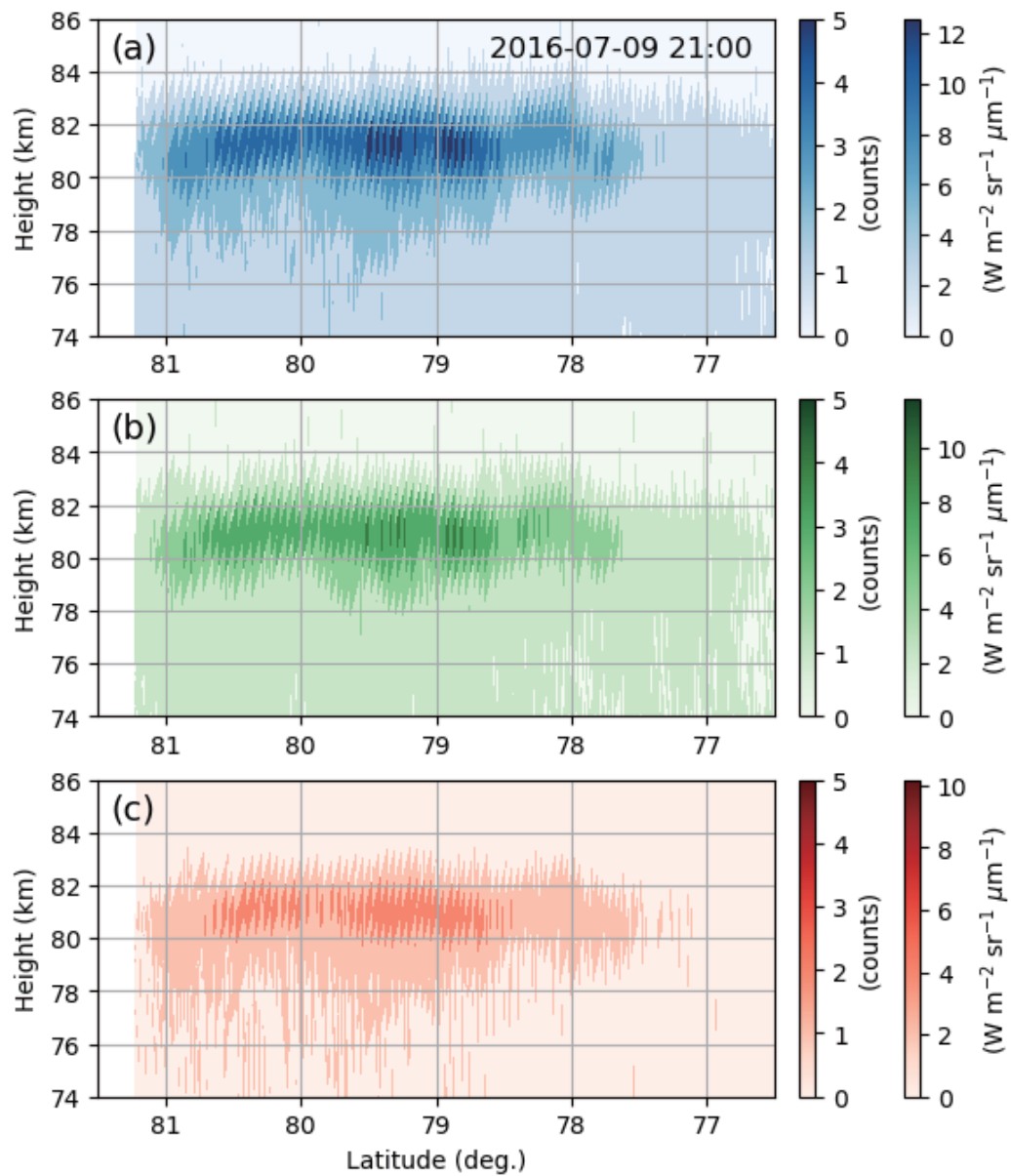

**Figure 2.** (a) Height-latitude distribution of emission intensity in the blue band at 21:00 UT on 9 July 2016. (b) Same as Figure 2a, but in the green band. (c) Same as Figure 2a, but in the red band.

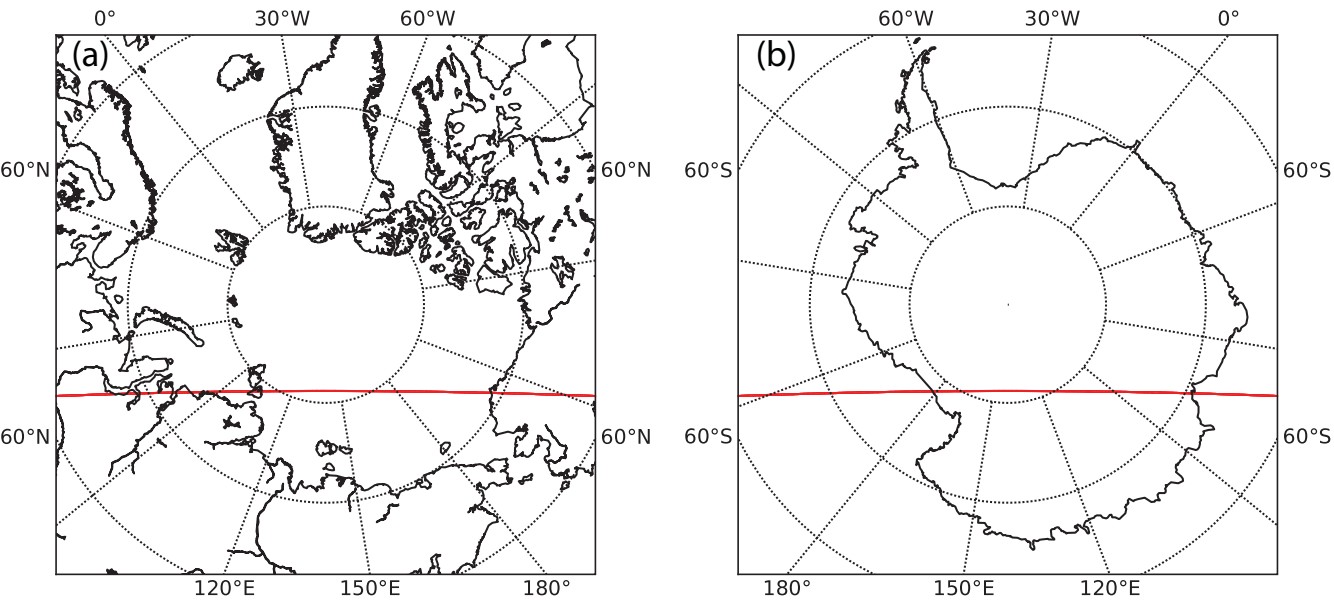

**Figure 3.** (a) Arctic map showing footprints (red line) of the tangential points at tangential heights of 80–85 km. (b) Same as Figure 3a, but Antarctic map. It should be noted that the center longitude is the sub-satellite longitude of Himawari-8: 140.7°E.

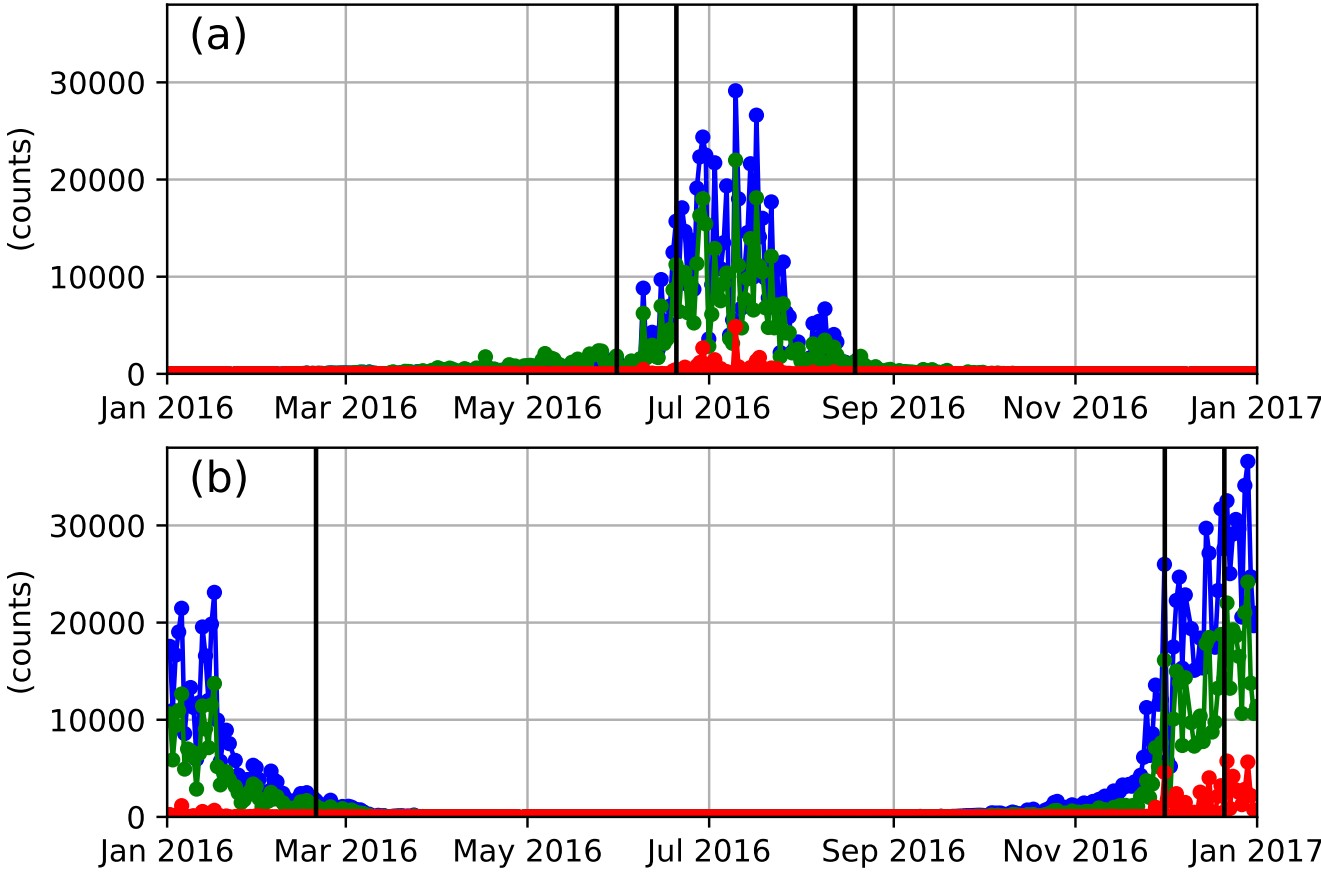

**Figure 4.** (a) Yearly variation in the total emission intensities for a region at heights of 70–90 km and latitudes of 60–90°N in 2016. Blue, green, and red lines correspond to the blue, green, and red visible bands, respectively. It should be noted that the upper limit of the latitude range is actually ∼81°, that is, the highest latitude of the tangential points when the height range is set to 70–90 km. Vertical black lines indicate the northern summer solstice in 2016 (i.e., 20 June 2016), 20 days before the solstice (i.e., 31 May 2016), and 60 days after the solstice (i.e., 19 August 2016), in reference to the typical PMC period. (b) Same as Figure 4a, but for a region at heights of 70–90 km and latitudes of 60–90°S. Vertical black lines indicate the southern summer solstice in 2016 (i.e., 21 December 2016), 20 days before the solstice (i.e., 1 December 2016), and 60 days after the 2015 southern summer solstice (i.e., 20 February 2016), in reference to the typical PMC period.