# Peer review of "Initial report on polar mesospheric cloud observations by Himawari-8"

_Atmospheric Measurement Techniques, 2018_

## Referee Comment (RC1) · Anonymous Referee #1 · 29 Jun 2018

**Review of Tsuda et al., "Polar mesospheric clouds observed by Himawari-8"**

This paper introduced a new satellite measurement of PMCs from Himawari-8, a limb viewing sensor on a geostationary satellite. It demonstrates clearly that the clouds seen are PMCs, depending on its altitudes (80-82 km), and seasonality in both northern and southern polar regions. It is a valid measurement paper and fairly well written. I recommend its publication at AMT with minor revisions.

Overall, there are a number of grammar mistakes throughout the manuscript.

Minor comments:

Page 1:
Line 3: "evidence"

Line 4: which 3 visible bands?

Line 5: suggest "contribute to the PMC research"

Line 14: suggest "CO2 cooling increase and more H2O"

Line 19: remove "from" after limb-reviewing

Line 20: suggest "a GEO satellite"

Page 2: line 5:  replace "by" with "from"

Page 3: line 1: "details"

Line 5: "this kind of"

Line 9: one tenth of the wavelength of what?

Line 10: This part of discussion is not clear. shorter wavelength light has stronger Mie scattering. Check the design of AIM CIPS and SOFIE.

Line 15: what's relation between total counts and emission intensities? Linear?

Line 23: define "sun-induced noise"

Figure 2: "at 21:00 UT"

---

## Referee Comment (RC2) · Y. Murayama (Referee) · 15 Jul 2018

Review "Polar mesospheric clouds observed by Himawari-8" by Tsuda et al.,

This paper has firstly employed high-resolution images observed with the meteorological satellite Himawari-8 to retrieve signals of polar mesospheric clouds (PMCs). Although this is not the first study using geostationary satellite images for PMC detection, the reviewer recognizes its important potential toward future development of PMC/NLC/PMSE research to understand detailed and long-term behaviors of PMCs by using the high-quality images obtained by such an operational meteorological satellite at the geostationary orbit. However, the reviewer suggests the weakness of this paper as follows; although it is the first report from Himawari-8 data of PMC observation, it is not an intensive validation study, nor it is not a strong message to show readers a big (more concrete) potential of this Himawari-8 data. The authors may wish to improve the manuscript from one of (or those multiple) viewpoints. The reviewer recommends publication after revising the manuscript regarding the comments below.

p.2, Section 2. Data or in any appropriate place: May the authors describe what longitudinal region the observation covers?

p.3, l.3: "Some fine structure" may not be clear for readers. At least specify time and height ranges as well as time and vertical scales of the features that the authors intend to show to readers.

Figure 2: Many slant line structures are found in the time-height sections, but no description is given on those. The authors may wish to address if they are artifact by instrumentation or not, and also what mechanism generates. This will give readers more insights for how to interpret the diagram.

p.3, ll. 11-13: In spite of mentioning possible use of the 3 visible bands for future estimation of particle size, it is not very clear about if those bands with their precisions and band widths etc. are suitable enough for estimating expected PMC particle sizes and distribution. The authors may wish to discuss how much this observation can be (can be expected to be) an advantage for such advanced PMC parameter estimations (quantitative discussion is welcome, but if too difficult, at least qualitatively please).

p.3, l.25: The "consistency" of Figure 3 with past observations is derived only from a simple feature of "summer increase". Is it possible for the authors to make more detailed comparison or argument, for further confirmation? For example, can we say that the increase rate of PMC occurrence from spring to summer (and decease rate from summer to fall) is consistent with other observations? Do a peak time and width of the summer-peak coincide ones in past observations? Or, the summer peak in the southern hemisphere is higher than one in the northern hemisphere. How do the authors discuss this feature in context of validating the PMC detection (consistency

with past observation?), or, if it is difficult, is it possible to discuss that this feature is not suitable for data validation or scientific analysis of PMC occurrence distribution?

p.3 ll.27-28: The authors claim that summer-increase signal is "concrete evidence" of PMC detection, without a further validation study (including detailed quantitative comparison and perhaps discussion of possible errors). The final conclusion is suggested to be left for the next or future validation study, unless more quantitative/qualitative discussion is made. At least points suggested by the other comments in this report are recommended to be clear. (The reviewer expected that, this paper may be to say "yes, we are successful to detect possible PMC signals (very likely PMCs, so it is exciting!)", and readers may expect the next paper "we made a validation study and the result was OK". Then those papers will be referred by future papers of scientific analysis and new findings? Please correct this reviewers' perspective if there are mistakes.)

Figure 3 and the third paragraph on p.3: Data processing is unclear to derive each data point in Figure 3. Is each point a daily value? What does "total count" mean (emission count data were summed up for the latitudinal range?; what altitude?), is the count value be linearly interpretable to an emission intensity value? —

---

## Author Comment (AC2) · 10 Oct 2018

amt-2018-120, Response to Referee #2

Dear Dr. Yasuhiro Murayama,

thank you very much for your insightful and helpful comments. We carefully revised the paper according to your comments together with comments from other reviewers. In addition, we used a commercial English language editing service to correct English mistakes in the manuscript.

Comments:
This paper has firstly employed high-resolution images observed with the meteorological satellite Himawari-8 to retrieve signals of polar mesospheric clouds (PMCs). Although this is not the first study using geostationary satellite images for PMC detection, the reviewer recognizes its important potential toward future development of PMC/NLC/PMSE research to understand detailed and long-term behaviors of PMCs by using the high-quality images obtained by such an operational meteorological satellite at the geostationary orbit. However, the reviewer suggests the weakness of this paper as follows; although it is the first report from Himawari-8 data of PMC observation, it is not an intensive validation study, nor it is not a strong message to show readers a big (more concrete) potential of this Himawari-8 data. The authors may wish to improve the manuscript from one of (or those multiple) viewpoints. The reviewer recommends publication after revising the manuscript regarding the comments below.
Reply:
Thank you very much for your understanding of importance of this paper. As you pointed out, we realized that this paper does not include intensive validation. Although the present investigation is a brief validation, a major point of this paper is to make a *rapid* report in the PMC observation by Himawari-8 to the related community. This is because there was no PMC report from the Japanese GEO satellite observations before this work. More concreate investigations are obviously needed, but it will take much more time. So, it is beyond the scope of the present paper. To make this point more clearly, we slightly rewrote title, abstract, the last paragraph in section 3, and summary.

Comments:

p.2, Section 2. Data or in any appropriate place: May the authors describe what longitudinal region the observation covers?

Reply:

To show the observation coverage, we added a figure (i.e., Figure 3).

Comments:

p.3, l.3: "Some fine structure" may not be clear for readers. At least specify time and height ranges as well as time and vertical scales of the features that the authors intend to show to readers.

Reply:

To clarify it, we rewrote the descriptions as follows.

"(3rd paragraph, section 3) We confirm that the PMC-like emission layer has a wavelike structure in the height-latitude cross section. For example, the heights of the layer were 80–81 km at ~81°N, 81–82 km at ~80°N, ~81 km at ~79°N, and 81-82 km at ~78°N. We attribute these fluctuations to atmospheric waves. Because the latitude range of 79–81°N corresponds to about a distance of ~700 km over the Arctic ocean (see Figure 3), the wavelength of the wavelike structure can be estimated to be ~700 km."

Comments:

Figure 2: Many slant line structures are found in the time-height sections, but no description is given on those. The authors may wish to address if they are artifact by instrumentation or not, and also what mechanism generates. This will give readers more insights for how to interpret the diagram.

Reply:

To clarify it, we rewrote the descriptions as follows.

"(2nd paragraph, section 3) Our observations also show oblique, step-like structures with a spatial scale of ~1 km. These artifactual structures are because of the height resolution and the mapping calculation of the tangential point. Before application of the mapping calculation, images show step-like structures that are not oblique but horizontal, with a spatial scale of ~1 km, because of ~1-km height or spatial resolution. These artifactual structures are because of a limitation of the height resolution. Such horizontal, step-like structures then become oblique, step-like structures through the mapping calculation from the pixel coordinate in the original images to the height-latitude coordinate in the tangential points."

Comments:

p.3, ll. 11-13: In spite of mentioning possible use of the 3 visible bands for future estimation of particle size, it is not very clear about if those bands with their precisions and band widths etc. are suitable enough for estimating expected PMC particle sizes and distribution. The authors may wish to discuss how much this observation can be (can be expected to be) an advantage for such advanced PMC parameter estimations (quantitative discussion is welcome, but if too difficult, at least qualitatively please).

Reply:

At this moment, it is not easy to give quantitative discussion, because this issue is beyond the scope of this paper. So, we rewrote the descriptions as follows. "(4th paragraph, section 3) Emission intensity was the strongest in the blue band, and the weakest in the red band. This result can be explained by Rayleigh or Mie scattering of sunlight by water-ice particles (i.e., PMCs). According to the Mie theory (cf. Bohren and Huffman, 2007), Rayleigh scattering is Mie scattering in the limit where particle size is much smaller than the wavelength. In the Rayleigh scattering region, the scattering cross section at a fixed wavelength decreases with decreasing particle size, and the scattering cross section at a fixed particle size decreases with increasing wavelength. Most PMC particles are understood to have radii of 20-60 nm, along with a very small number of ~200 nm particles (cf. DeLand et al., 2006). Such particle size is smaller than the visible observation wavelengths: blue (0.47 μm), green (0.51 μm), and red (0.64 μm). Hence, the scattering would be close to Rayleigh scattering, whereas it may not be the pure Rayleigh scattering. Therefore, the predominance of such small particles (20-60 nm) would account for the higher scattering intensities or stronger emission intensities at shorter wavelengths. Although additional quantitative evaluation would be helpful, the inferential framework described above would imply that the three visible images obtained by Himawari-8 may contain information on the size of PMC particles. In addition, Himawari-8 observation can cover an entire day of local time (i.e., 00:00-24:00 JST), which means that emission data at almost all scattering angles (i.e., 0-360°) are available. Such information may be useful for particle size investigation, considering the wavelength dependence and the angle distribution in the Rayleigh to Mie scattering region. However, this concern is beyond the scope of the present paper."

Comments:

p.3, l.25: The "consistency" of Figure 3 with past observations is derived only from a simple feature of "summer increase". Is it possible for the authors to make more detailed comparison or argument, for further confirmation? For

example, can we say that the increase rate of PMC occurrence from spring to summer (and decease rate from summer to fall) is consistent with other observations? Do a peak time and width of the summer-peak coincide ones in past observations? Or, the summer peak in the southern hemisphere is higher than one in the northern hemisphere. How do the authors discuss this feature in context of validating the PMC detection (consistency with past observation?), or, if it is difficult, is it possible to discuss that this feature is not suitable for data validation or scientific analysis of PMC occurrence distribution?

Reply:

To make a more careful investigation, we rewrote the descriptions as follows. "(5th paragraph, section 3) As shown in Figures 4a and 4b, PMC-like emissions, expressed as total pixel values, were active only during local summer months. The observed active periods would be similar to the typical PMC active period, from ~20 days before summer solstice to ~60 days after summer solstice (cf. DeLand et al., 2006)."

Comments:

p.3 ll.27-28: The authors claim that summer-increase signal is "concrete evidence" of PMC detection, without a further validation study (including detailed quantitative comparison and perhaps discussion of possible errors). The final conclusion is suggested to be left for the next or future validation study, unless more quantitative/qualitative discussion is made. At least points suggested by the other comments in this report are recommended to be clear. (The reviewer expected that, this paper may be to say "yes, we are successful to detect possible PMC signals (very likely PMCs, so it is exciting!)", and readers may expect the next paper "we made a validation study and the result was OK". Then those papers will be referred by future papers of scientific analysis and new findings? Please correct this reviewers' perspective if there are mistakes.)

Reply:

As you pointed out, we realized that this paper does not include intensive validation. Although the present investigation is a brief validation, a major point of this paper is to make a *rapid* report in the PMC observation by Himawari-8 to the related community. This is because there was no PMC report from the Japanese GEO satellite observations before this work. More concreate investigations are obviously needed, but it will take much more time. So, it is beyond the scope of the present paper. To make this point more clearly, we slightly rewrote title, abstract, the last paragraph in section 3, and summary.

Comments:

Figure 3 and the third paragraph on p.3: Data processing is unclear to derive each data point in Figure 3. Is each point a daily value? What does "total count" mean (emission count data were summed up for the latitudinal range?; what altitude?), is the count value be linearly interpretable to an emission intensity value?

Reply:

To clarify it, we rewrote some descriptions as follows.

[revised manuscript text omitted]

---

## Author Comment (AC1)

Dear Referee,

thank you very much for your insightful and helpful comments. We carefully revised the paper according to your comments together with comments from other reviewers. In addition, we used a commercial English language editing service to correct English mistakes in the manuscript.

Comments:
This paper introduced a new satellite measurement of PMCs from Himawari-8, a limb viewing sensor on a geostationary satellite. It demonstrates clearly that the clouds seen are PMCs, depending on its altitudes (80-82 km), and seasonality in both northern and southern polar regions. It is a valid measurement paper and fairly well written. I recommend its publication at AMT with minor revisions. Overall, there are a number of grammar mistakes throughout the manuscript.
Reply:
Thank you very much for your understanding of importance of this paper.

Comments:
Minor comments:
Page 1:
Line 3: "evidence"
Reply:
The mistake was removed during the revision.

Comments:
Line 4: which 3 visible bands?
Reply:
To clarify it, we rewrote the descriptions as follows.
"(abstract) three visible bands: blue (0.47 μm), green (0.51 μm), and red (0.64 μm)"

Comments:
Line 5: suggest "contribute to the PMC research"
Reply:
Modified as suggested.

Comments:
Line 14: suggest "CO2 cooling increase and more H2O"
Reply:
Modified as "CO2 cooling and H2O increase".

Comments:
Line 19: remove "from" after limb-reviewing
Reply:
Fixed as suggested.

Comments:
Line 20: suggest "a GEO satellite"
Reply:
Modified as suggested.

Comments:
Page 2: line 5: replace "by" with "from"
Reply:
Fixed as suggested.

Comments:
Page 3: line 1: "details"
Reply:
Fixed as suggested.

Comments:
Line 5: "this kind of"
Reply:
The mistake was removed during the revision.

Comments:

Line 9: one tenth of the wavelength of what?

Reply:

As you pointed out, the descriptions were not clear. So, we rewrote the descriptions as follows.

"(4th paragraph, section 3) Emission intensity was the strongest in the blue band, and the weakest in the red band. This result can be explained by Rayleigh or Mie scattering of sunlight by water-ice particles (i.e., PMCs). According to the Mie theory (cf. Bohren and Huffman, 2007), Rayleigh scattering is Mie scattering in the limit where particle size is much smaller than the wavelength. In the Rayleigh scattering region, the scattering cross section at a fixed wavelength decreases with decreasing particle size, and the scattering cross section at a fixed particle size decreases with increasing wavelength. Most PMC particles are understood to have radii of 20-60 nm, along with a very small number of ~200 nm particles (cf. DeLand et al., 2006). Such particle size is smaller than the visible observation wavelengths: blue (0.47 μm), green (0.51 μm), and red (0.64 μm). Hence, the scattering would be close to Rayleigh scattering, whereas it may not be the pure Rayleigh scattering. Therefore, the predominance of such small particles (20-60 nm) would account for the higher scattering intensities or stronger emission intensities at shorter wavelengths. Although additional quantitative evaluation would be helpful, the inferential framework described above would imply that the three visible images obtained by Himawari-8 may contain information on the size of PMC particles. In addition, Himawari-8 observation can cover an entire day of local time (i.e., 00:00-24:00 JST), which means that emission data at almost all scattering angles (i.e., 0-360°) are available. Such information may be useful for particle size investigation, considering the wavelength dependence and the angle distribution in the Rayleigh to Mie scattering region. However, this concern is beyond the scope of the present paper."

Comments:

Line 10: This part of discussion is not clear. shorter wavelength light has stronger Mie scattering. Check the design of AIM CIPS and SOFIE.

Reply:

As you pointed out, the descriptions were not clear. So, we rewrote the descriptions in 4th paragraph, section 3 (see the reply above).

Comments:
Line 15: what's relation between total counts and emission intensities? Linear?
Reply:
To clarify it, we rewrote some descriptions as follows.
"(section 2) The color value has a linear relation with the emission intensity for each band."
"(5th paragraph, section 3) To investigate seasonal variation of PMC emissions in 2016, we calculated total pixel values in specific regions (i.e., sum of pixel values in localized pixels)."

Comments:
Line 23: define "sun-induced noise"
Reply:
To clarify it, we rewrote the descriptions as follows.
"(5th paragraph, section 3) This configuration would be beneficial, providing solar illumination to some extent while minimizing sun-induced noise, such as the stray light of direct sunlight, which can be a problem close to local midnight (00:00 JST) (cf. Proud, 2015).

Comments:
Figure 2: "at 21:00 UT"
Reply:
Fixed as suggested.

[revised manuscript text omitted]